

# Quantifying uncertainties of permafrost carbon-climate feedbacks

Eleanor J Burke[1], Altug Ekici[2,3], Ye Huang[4], Sarah E Chadburn[2,5], Chris Huntingford[6], Philippe Ciais[4], Pierre Friedlingstein[2], Shushi Peng[4,7] and Gerhard Krinner[8]

[1]Met Office Hadley Centre, FitzRoy Road, Exeter, EX1 3PB, UK
[2]University of Exeter, College of Engineering, Mathematics and Physical Sciences, Exeter, EX4 4QF, UK
[3]Uni Research Climate and Bjerknes Centre for Climate Research, Bergen, Norway
[4]Laboratoire des Sciences du Climat et de l'Environnement, UMR 1572 CEA-CNRS-UVSQ, Gif sur Yvette 91191, France
[5]University of Leeds, School of Earth and Environment, Leeds, LS2 9JT, UK
[6]Centre for Ecology and Hydrology, Wallingford, Oxfordshire, OX10 8BB, U.K.
[7]Sino-French Institute for Earth System Science, College of Urban and Environmental Sciences, Peking University, Beijing 100871, China.
[8]Laboratoire de Glaciologie et Géophysique de l'Environnement, 54 rue Molière, F-38402, Saint Martin d'Hères, France.

*Correspondence to*: Eleanor J. Burke (eleanor.burke@metoffice.gov.uk)

**Abstract.** The land surface models JULES (two versions) and ORCHIDEE-MICT, each with a revised representation of permafrost carbon, were coupled to the IMOGEN intermediate complexity climate and ocean carbon uptake model. IMOGEN calculates atmospheric carbon dioxide ($CO_2$) and local monthly surface climate for a given emission scenario with the land-atmosphere $CO_2$ flux exchange from either JULES or ORCHIDEE-MICT. These simulations include feedbacks associated with permafrost carbon
changes in a warming world. Both IMOGEN-JULES and IMOGEN-ORCHIDEE-MICT were forced by historical and three alternative future $CO_2$ emission scenarios. Simulations were performed for different climate sensitivities and regional climate change patterns based on 22 different Earth System Models (ESM) used for CMIP3 (phase 3 of the Coupled Model Intercomparison Project), allowing us to explore climate uncertainties in the context of permafrost carbon − climate feedbacks. Three future emission
scenarios consistent with three representative concentration pathways: RCP2.6; RCP4.5 and RCP8.5 were used. Paired simulations with and without frozen carbon processes were required to quantify the impact of the permafrost carbon feedback on climate change. The additional warming from the permafrost carbon feedback is between 0.2 and 12 % of the change in the global mean temperature ($\Delta T$) by year 2100 and 0.5 and 17 % of $\Delta T$ by 2300, this range reflecting differences in land surface models,
climate models and emissions pathway. As a percentage of $\Delta T$, the permafrost carbon feedback has a greater impact on the low emission scenario (RCP2.6) than on the higher emissions scenarios





suggesting that permafrost carbon should be taken into account when evaluating heavy mitigation and stabilizations scenarios. Structural differences between the land surface models are found to be a larger source of uncertainties than differences between climate models, in particular due to different representations of soil carbon decomposition. Inertia in the permafrost carbon system means that the

permafrost carbon response is dependent on the temporal trajectory of warming as well as the absolute amount of warming. We propose a new policy relevant metric - the Frozen Carbon Vulnerability timescale (FCVt) in years - that can be derived from the more complex land surface models and used to quantify the permafrost carbon response given any pathway of global temperature change.

## 1 Introduction

The coupling between the global carbon cycle and the rest of the climate system gives rise to a range of feedbacks to climate on multiple time scales. These feedbacks are expressed in the future by either amplifying or mitigating any change implied by a given fossil fuel and cement production emission scenario. They are highly uncertain. For example, Jones et al. (2013) showed that inter-model uncertainty in the projected change in land carbon uptake of atmospheric $CO_2$ over the 21[st] century is

comparable with the implications, on atmospheric $CO_2$, of the spread across emission scenarios. In addition Earth System Models (ESMs) do not represent all of the relevant feedbacks. In the northern high latitudes, the new generation of climate models in the Coupled Model Intercomparison Project Phase 5 (CMIP5) ensemble simulate a warming-induced uptake of carbon, although with a low confidence (Ciais et al., 2013). However, none of these CMIP5 models include a representation of the

large stocks of 'old' permafrost carbon which, although today is stabilised by frozen and/or by saturated conditions may become active and release $CO_2$ or $CH_4$ under global warming (Hugelius et al., 2014; Gorham, 1991). Adding the permafrost-carbon response to climate may change the CMIP5 model simulations of the northern high latitudes from a sink to a source of carbon and thus a positive feedback (Burke et al., 2013; Koven et al., 2011; Ciais et al., 2013). For this reason permafrost processes must be

routinely included in the simulations of the global carbon cycle.



Estimates of the impact of climate change on permafrost carbon have typically been performed combining estimates of soil thermal changes with those of simplified soil carbon decomposition (Burke et al., 2012; Koven et al., 2015; Schneider von Deimling et al., 2015). Schuur et al. (2015) collated results from many of these studies and showed that the potential carbon release from today's permafrost zone is between 37 and 174 Gt carbon by the year 2100 and under a "business-as-usual" scenario (Representative Concentration Pathway RCP 8.5; Meinshausen et al., 2011). This is comparable with the later result of Koven et al. (2015) who estimated a permafrost carbon response of 28–113 Gt C for the same time period and scenario based on a soil carbon decomposition model in which the response of soil carbon to warming was calibrated by the results of laboratory incubation experiments (Schädel et al. 2014).

The response of the land carbon cycle to climate change can be separated into two different components - its response to $CO_2$ and its response to climate approximated by global mean warming (Friedlingstein et al., 2006). The carbon - climate feedback parameter, $\gamma$, defined using the CMIP5 models without permafrost ranges from a release of 16 to 89 Gt C $^O$C$^{-1}$ from the land surface (Arora et al., 2013). For the CMIP5 models, this is offset by $CO_2$ fertilisation of the land surface, making the land surface a net sink. Burke et al. (2013) estimated the permafrost-specific carbon feedback ($\gamma_{PF}$) that was missing in CMIP5 models, i.e. the relationship between the release of carbon from permafrost soils and global temperature change. They estimated $\gamma_{PF}$ at 2100 to range from an additional release of 6 to 66 Gt C $^O$C$^{-1}$. This is of comparable magnitude to all the other land carbon feedbacks and could change the overall land surface to become a net source of carbon. MacDougall (2016) used a permafrost carbon-enabled intermediate complexity climate model and confirmed the large magnitude of $\gamma_{PF}$ but also showed that $\gamma_{PF}$ increases significantly over time from around 24 Gt C $^O$C$^{-1}$ in 2100, to around 47 Gt C $^O$C$^{-1}$ in 2300. This suggests that $\gamma_{PF}$ could be pathway and time dependent and the linear feedback theory developed by Friedlingstein et al. (2003) is not valid when incorporating the response of permafrost carbon to warming.





The additional permafrost release of carbon to the atmosphere amplifies global warming forced by anthropogenic emissions, and the amount of permafrost carbon released under various emission scenarios and at different time scales has been estimated in a range of studies (e.g. Schaefer et al., 2011, Koven et al. 2011; 2015). However, there are currently only a few estimates of the impact of this feedback in terms of additional climate change. Burke et al. (2013) and Schneider von Deimling et al (2012; 2015) used a simple climate energy balance model to show the temperature amplification of the permafrost carbon feedback is between 0.02 and 0.36 $^{\mathrm{O}}$C by 2100. MacDougall et al. (2012; 2013) found that including permafrost carbon within their intermediate complexity climate model increased the global mean temperature by an additional 0.1 and 0.8 $^{\mathrm{O}}$C by 2100. They found the permafrost carbon released under low emission scenarios provides a more significant climate feedback than the permafrost carbon released under high emission scenarios. Indeed a kilogram of $CO_2$ transferred to the atmosphere under a low emissions pathway has a higher radiative efficiency than the same kilogram of $CO_2$ released under a high emissions pathway. In the MacDougall et al. (2012) study this factor outweighs the more limited permafrost carbon loss at lower emissions. Similarly, using the CLIMBER-2 intermediate complexity climate model, Crichton et al. (2016) suggest a relative increase of peak temperature change between 10 and 40 %, depending on the emission scenario, with RCP4.5 being most affected.

To explore sources of uncertainty in these estimates, we use a coupled climate modelling system of intermediate complexity with new generation process-oriented land surface models including permafrost processes. This framework allows us to make a more comprehensive assessment of the permafrost carbon response to climate change and its subsequent impact on global temperature including a wide spectrum of uncertainties of future emissions scenario (policy uncertainty); climate response to increased radiative forcing (climate sensitivity and regional distribution of climate change); and parameterisation of the soil carbon decomposition (terrestrial process uncertainty). Three different versions of global land surface schemes (JULES-Resp$_{deep}$; JULES-suppressR$_{esp}$; and ORCHIDEE-MICT) are coupled with the IMOGEN intermediate complexity climate model (Huntingford et al., 2010). IMOGEN was tuned to represent the response of 22 available GCMs from CMIP3 (phase 3 of

the Coupled Model Intercomparison Project). [The range of climate sensitivity and regional distribution of climate change in the CMIP3 models is comparable with that in the CMIP5 models.] IMOGEN was driven out to 2300 using harmonised emissions scenarios corresponding to RCP2.6, RCP4.5 and RCP8.5 (Meinshausen et al., 2011). This work therefore provides a rigorous assessment of the uncertainty range of the permafrost climate-carbon feedbacks using land surface components representative of the next generation of Earth System Models that will be used for the upcoming IPCC assessment.

## 2 Materials and Methods

### 2.1 JULES land surface scheme

The Joint UK Land Environment Simulator (JULES - Best et al., 2011; Clark et al., 2011) is the land surface component of the UK Earth System Model (UKESM - Jones and Sellar, 2016). This paper uses a permafrost-adapted version of JULES (version 4.3; Chadburn et al., 2015a). JULES describes the physical, biophysical and biochemical processes that control the exchange of radiation, momentum, heat, water and carbon between the land surface and the atmosphere. It can be applied at a point or over a grid, and requires temporally continuous meteorological forcing data along with atmospheric $CO_2$ concentration. Each point or grid box can contain several different land-cover types or "tiles", including five plant functional types (broadleaf trees, evergreen trees, $C_3$ and $C_4$ grasses and shrubs) as well as non-vegetated tiles (urban, water, ice and bare soil). Each tile has its own surface energy balance, but the soil underneath is treated as a single column and receives aggregated mean fluxes from the surface tiles. TRIFFID, the dynamic vegetation model (Clark et al., 2011), was used to simulate the vegetation distribution and its response in a changing climate.

Several new modifications have been added into JULES to improve the representation of physical and biogeochemical processes in the cold regions. These include the additional impact of the insulation effects of a fractional moss layer at the soil surface; updated soil thermal and hydraulic properties to take account of the presence of organic matter; and a deeper and better resolved soil column, with an





additional thermal column at the base of the soil to represent bedrock (Chadburn et al., 2015a; 2015b). These changes lead to a significant reduction of the error in the annual cycle of soil temperature along with a reduction in the active layer bias, from over 1.0 m too deep to only about 0.4 m too deep. All these developments are included here in an improved JULES version better suited for the permafrost

simulations discussed here.

The standard soil carbon model in JULES is a 4-pool model (decomposable plant material, resistant plant material, biomass and humus). When added together these pools represent the total soil carbon storage. The model is based on the RothC soil carbon model and described in detail in Clark et al.

(2011). Burke et al. (2016) adapted the soil carbon model in JULES to include a soil vertical dimension within each of the carbon pools. This results in a set of pools in every layer of the soil column. The respiration rate is determined at each depth ($z$) for each soil carbon pool ($i$) and is given by:

$$R_i = k_i C_i(z) F_T(T_{soil}(z)) F_s(s(z)) F_v(v) exp(-z/\zeta_{resp}) \qquad (1)$$

Here $k_i$ is a pool specific decay constant (s$^{-1}$), $C_i$ is the amount of soil carbon in pool $i$ (kg m$^{-2}$), $F_T$, $F_s$ and $F_v$ parameterise the response of the respiration rate to temperature ($T_{soil}(z)$ in K), soil moisture ($s(z)$ as a fraction of saturation) and vegetation fraction ($v$) respectively. The soil respiration is additionally modified by including an extra exponential decay of respiration with depth. This accounts for factors

that are currently missing in the model such as priming effects and microscale anoxia (Koven et al., 2013). The e-folding depth ($\zeta_{resp}$ in m) of this function is very uncertain and the soil carbon vertical distribution depends significantly on its value (Burke et al., 2016). A smaller $\zeta_{resp}$ means the respiration is more suppressed with depth and results in more soil carbon particularly in the deeper soils.

Two different parameterisations of the response of respiration to temperature ($F_T$) are available within JULES (Clark et al., 2011) and we test both. JULES-suppressR$_{esp}$ uses an Arrhenius function ($F_{T,Q10}$ from Equation 2) with $Q_{10}$=2.0 and $\zeta_{resp}$ = 0.56 m, whereas JULES-deepR$_{esp}$ uses $F_{T,Roth}$ in Equation 3 and $\zeta_{resp}$ = 2.5 m.



$$F_{T,Q10}(T_{soil}) = Q_{10}^{\frac{T_{soil}-298.15}{10}} \tag{2}$$

$$F_{T,Roth}(T_{soil}) = 47.91 + \exp\left(\frac{106.0}{T_{soil}-254.85}\right) \tag{3}$$

Burke et al. (2016) showed there was very little difference in the timing of the peak soil respiration in summer between these two temperature response functions when combined with appropriate e-folding depths ($\zeta_{resp}$).

Soil carbon increases though vegetation litter fall. Although the majority of the litter enters at the soil surface, a small amount enters the deeper soil layers, for example, from roots. In JULES the litter distribution drops off exponentially with depth with an e-folding parameter of 5 m$^{-1}$. The amount and quality of litter directly impacts the soil carbon stocks, therefore it is important for the simulated vegetation distribution to be as accurate as possible.

There is a vertical mixing term representing either bioturbation (i.e. the soil mixing by, for example, animals and plant roots), or, in permafrost regions, cryoturbation (soil mixing is from frost heave and freeze-thaw processes). The mixing rate changes depending on whether permafrost is present or not (Burke et al., 2016, Koven et al., 2013). In the absence of permafrost, the bioturbation mixing rate is constant at 1 cm$^2$ year$^{-1}$. The cryoturbation mixing rate is set at 5 cm$^2$ year$^{-1}$. This drops off linearly below one meter, reaching zero at 3 m depth. Permafrost is diagnosed at any location where the deepest soil layer is below 0 $^{O}$C, assuming that there is only a very minor seasonal cycle in temperature at this depth.

Unique to the analysis is that in JULES, a tracer was added to enable the 'old carbon' initially within the permanently frozen soils to be easily distinguished from the rest of the soil carbon (Burke et al., 2016). This enables the old permafrost carbon, defined as carbon within the permanently frozen soil at





the start of the simulation, to be traced throughout the simulation. It is this 'old carbon' that thaws under global warming, that we wish to characterize in any additional contribution of land-atmospheric $CO_2$ flux. It is the additional contribution to the land-atmospheric CO2 flux of this 'old carbon' that becomes thawed under global warming, that we wish to quantify.

## 2.2 ORCHIDEE-MICT

Our second land surface model is the ORCHIDEE-MICT model, again enhanced with several new processes related to cold region soils. The new soil processes include the implementation of the thermal and hydrological effects of soil freezing in a multi-layered soil hydrology scheme (Gouttevin et al., 2012). Gouttevin et al. (2012) state that the modelling of the soil thermal regime is generally improved by the representation of soil freezing processes. This enables the dynamics of the active layer to be more accurately captured. This process is important when simulating the response of frozen carbon stocks to future warming (Koven et al., 2009; 2011). Also added is a more advanced multi-layer snow scheme, which improves the estimation of permafrost physics (Wang et al., 2013). This three-layered snow module includes a varying snow density and a varying snow thermal conductivity along with the thawing and refreezing of water within the snowpack. More specifically, the snow module has been introduced to account for the water freezing/thawing processes within snow capturing more accurately the impact of the overlying snow cover on soil temperature (Wang et al., 2013). An evaluation of snow depth, snow water equivalent, surface temperature, snow albedo, and snowmelt runoff demonstrate the improvement in the simulation of snow processes by this version of ORCHIDEE-MICT over previous versions. To account for the effects of cryoturbation on redistribution of soil organic carbon (SOC), a vertical mixing scheme based on a diffusion equation was introduced into ORCHIDEE-MICT (Koven et al., 2009), with the diffusion length being set to 3 times the local active layer thickness. In the model version used here, carbon and temperature are discretised down to the depth of the bottom layer (47.6 m), whereas the soil depth for hydrology is 2 m. Soil water content in each layer below 2 m is assumed to be equal to the monthly average soil moisture at the bottom layer of the top 2 m, and its frozen fraction depends on soil temperature of the layer below 2 m.





The soil carbon model of ORCHIDEE-MICT is based on the equations in the CENTURY model (Parton et al. 1992). It contains 7 pools, namely: above- and below-ground metabolic and structural litter, along with active, slow and passive soil organic carbon pools. Decomposition of carbon is modulated by soil temperature and moisture functions along with a clay function. Transfer functions between pools are described using the CENTURY equations (Parton et al. 1992). The temperature function $F_T$ follows equation (2) for temperatures above 0 °C. At colder soil temperatures below 0 °C, $F_T$ is reduced linearly decrease to reach zero at -1 °C (Koven et al. 2011). In this paper, heat production by decomposing soil carbon (the 'heating' experiment in Koven et al. 2011) is turned off. Unlike JULES the 'old carbon' cannot be traced throughout the simulation which means the 'old carbon' below the active layer and within the permafrost is only identified at the start of the simulation. As with JULES, the dynamic vegetation model was used to simulate the vegetation distribution and litterfall. Both of these have a significant impact on the soil carbon stocks.

## 2.3 IMOGEN

The Integrated Model Of Global Effects of climatic aNomalies (IMOGEN) is an intermediate complexity climate model developed specifically to quantify geographical and seasonal variation in meteorological conditions over land in response to changing atmospheric gas composition. It can be operated for different anthropogenic emission scenarios, and can capture global land-atmosphere carbon feedbacks. IMOGEN is calibrated to emulate different GCMs and, for example, it has recently been used to investigate the risk of Amazon dieback under a large range of climate projections (Huntingford et al., 2013). Here it provides a test bed for evaluating the impact of the permafrost feedback on the global carbon cycle for a variety of emission scenarios, driving GCMs and alternative land surface parameterisations describing the northern latitude terrestrial cryosphere response.

IMOGEN contains a simple energy balance model to relate changes in atmospheric greenhouse gas concentrations to the global mean land temperature via changes in a radiative forcing. The radiative forcing is itself dependent on any pathway in altered atmospheric gas concentrations since the pre-industrial period. The Energy Balance Model (EBM) requires four parameters which are readily





calibrated against given climate model (Huntingford et al., 2010). The four parameters are: climate feedback parameters over land and over sea, the oceanic effective thermal diffusivity representing the ocean thermal inertia and a land-sea temperature contrast parameter which linearly relates warming over the land to warming over the ocean (Huntingford and Cox, 2000).

IMOGEN forces its coupled land surface model with local meteorological data temporally downscaled from calculated mean monthly values, to 30 minute timescales using a weather generator. These driving data, required by both JULES and ORCHIEE-MICT, are 1.5m temperature, relative humidity, wind speed, precipitation, downward shortwave and longwave radiation and pressure. The mean monthly data
(that is downscaled) are derived for each GCM, assuming simple linear regressions between the local and monthly variations in meteorology and the amount of annual global mean warming over land. This "pattern-scaling" concept (Huntingford and Cox, 2000) takes these regression values, and multiplies them by the mean warming over land calculated from the EBM in IMOGEN. The patterns of changing meteorological conditions plus the four energy balance model parameters to give mean land warming
were calibrated for the 22 CMIP3 climate models (Huntingford et al., 2013). These 22 patterns represent the uncertainty in the driving climate models. The monthly anomalies of climate change (from EBM and patterns combined) are added to the 1961-1990 WATCH climatology (Weedon et al. 2011), which is assumed here to be also representative of pre-industrial conditions. Any biases introduced by neglecting anthropogenically induced climate change up to that date are assumed to be small compared
with the errors from using earlier years in the WATCH climatology with poorer observational coverage (Huntingford et al., 2013). This also removes individual GCM biases in the estimation of the pre-industrial state.

IMOGEN has a closed global carbon cycle when its operation includes a land surface model
(Huntingford et al., 2013). At the end of each modelled year, atmospheric $CO_2$ concentration is modified using the difference between prescribed emissions and the global mean ocean-atmosphere and land-atmosphere fluxes of $CO_2$ for that year. The values of Net Ecosystem Productivity (NEP) are integrated over all land points for that year and used to derive the land-atmosphere flux. The NEP is





output from either JULES or ORCHIDEE-MICT. A single "box" model is used to calculate the ocean sink. It is a function of both global temperature increase and atmospheric $CO_2$ level (Huntingford et al., 2004). Any changes in atmospheric $CO_2$ concentration then feed back via the energy balance model on modelled surface climate changes, which drives the scaled patterns of local and monthly climatology.

## 2.4 Experimental design

The pre-industrial spun up state for each of the different land surface models was estimated using the 1961-1990 WATCH climatology and pre-industrial atmospheric $CO_2$ concentration at the IMOGEN resolution of 2.5 degrees latitude and 3.75 degrees longitude. This was done independently for each of the three different global land surface model configurations but, in each case it was sufficient to give stable soil carbon and vegetation carbon distributions for 1860. In both JULES and ORCHIDEE-MICT competition of vegetation was enabled allowing the models to determine both their initial vegetation distributions and litterfall and the response of the vegetation distribution and litterfall to climate change. Anthropogenic land use change was ignored in these simulations, as it is relatively small in the northern high latitudes (Klein Goldewijk , 2001).

In JULES a 'modified accelerated decomposition' numerical technique (Koven et al., 2013; modified-AD) was adopted to more quickly spin the JULES soil carbon to an initial equilibrium distribution. The decay rates of the four soil carbon pools were set to the rate of the fastest pool. In order to appropriately adopt the modified-AD method, the diffusion coefficients for the four pools were multiplied by the same factors. The model was then initially spun-up for 500 model years using this modified-AD technique and the fixed WATCH climatology representative of pre-industrial times. The decay rates for the four pools were then reset and the model spun up for another 2000 years again using the WATCH climatology. This needed to be done independently for both JULES-suppress$R_{esp}$ and JULES-deep$R_{esp}$ - although these two model versions have the same physics and vegetation carbon, they have different soil carbon distributions. ORCHIDEE-MICT was initially spun-up by running the full version of the land surface model (30 minute timestep) for 150 years first, again with the WATCH climatology. Following this, the soil carbon sub-model forced by above- and below-ground litter input

(FORCESOIL) was run 10000 model years for 10 times each time followed by 2 years run of the full ORCHIDEE-MICT. This was followed by another 200 years of ORCHIDEE-MICT to complete the numerical spinup. Note that due to its permanent burial of carbon below the active layer even after 100,000 years of spinup, ORCHIDE-MICT's soil carbon pools continue to gain carbon, but at a very small rate (mean Net Ecosystem Productivity over the last 50 years of spinup is 0.16 Gt C yr$^{-1}$). The permafrost area, soil and vegetation carbon distributions for these pre-industrial states are described here and used to initialise the transient simulations.

In order to quantify the permafrost carbon feedback separately, paired simulations were carried out for each of the JULES and two ORCHIDEE simulations; one which includes the response of the climate to the $CO_2$ emissions from the perturbed (thawing) permafrost carbon (indexed "PF") and one which excludes it (indexed "non-PF"). The permafrost carbon was defined as the carbon which is below the active layer, or in the permanently frozen soils at the start of the simulation, i.e. representative of pre-industrial times.

The spun-up coupled system is forced with historical fossil fuel and cement production $CO_2$ emissions followed by the emissions representing three of the Representative Concentration Pathways (RCPs) used in the fifth assessment report of the Intergovernmental Panel on Climate Change (IPCC AR5) - RCP2.6, RCP4.5 and RCP8.5 (Moss et al., 2010; Meinshausen et al., 2011). Simulations were carried out until year 2300 using the RCP extensions (Meinshausen et al., 2011) in order to examine the long-term relationship between permafrost and climate. Non-$CO_2$ greenhouse gases and aerosols were not included in this set of simulations, nor were land use change emissions. The impact of these extra emissions will be minor for the purpose of our study focusing on the differences between PF and non-PF simulations.





## 3 Results

### 3.1 Evaluation of models

The models were assessed to ensure that the permafrost physics and the soil and vegetation carbon are not inconsistent with the observations. Permafrost is assumed to exist in grid cells where the soil is

frozen at 3m depth for a period of 2 years or more. Figure 1 (left panels) shows the simulated permafrost extent for JULES [JULES-suppress$R_{esp}$ and JULES-deep$R_{esp}$ have the same physics and hence the same permafrost] and ORCHIDEE-MICT. Superimposed on the simulated permafrost extent are the observations from Brown et al. (1998). Both JULES and ORCHIDEE-MICT capture all of the observed continuous permafrost. They might be expected to also capture the regions of discontinuous

permafrost (more than 50 % of a grid cell underlain by permafrost) and simulate a permafrost area similar to the observed area of continuous and discontinuous permafrost (15 million km$^2$). JULES has slightly too much permafrost overall with extra permafrost in Eurasia and not enough in North America - this is possibly caused by biases in the winter snow depth. ORCHIDEE-MICT systematically simulates more permafrost than either the observations or JULES. Compared with the zero degree

isotherm for the 2 m air temperature (Figure 1: right panels), ORCHIDEE-MICT has some permafrost where the annual mean temperature is greater than zero suggesting it might be missing a process which increases the thermal insulation in winter between the air and the deeper soil.

The simulated vegetation carbon distribution is shown in Figure 2. There are no feedbacks from the soil

carbon onto the vegetation - via, for example, changing soil hydraulic properties or nitrogen limitation - therefore both versions of JULES also have the same vegetation distribution. In general both of the models simulate more vegetation carbon than observed which will lead to more litter carbon input. Some model overestimation might be expected because there is no land use change included in the models. There are also some differences in spatial patterns, for example, in JULES the simulated boreal

forest does not extend far enough east in Siberia. This will reduce the litter inputs in eastern Siberia and potentially result in relatively smaller simulated soil carbon stocks in these regions. ORCHIDEE-MICT has slightly more vegetation carbon than JULES, but its spatial distribution is more comparable to the observations.



Figure 3 shows the soil carbon distribution simulated by the three different model versions (top three rows) with the left hand panels being the total soil carbon in the top 2 m and the right-hand panels being the soil carbon in the permafrost in the top 3 m. Also shown, bottom row, are two different observational data sets. The first is the ISRIC-WISE derived soil property estimates on a 30 by 30 arcsec global grid (WISE30sec; Batjes, 2016). The second is the Northern Circumpolar Soil Carbon Database version 2 (NCSCDv2; Hugelius et al., 2014). The WISE30sec soil carbon distribution for the top 2m of soil is shown at the bottom left and the NCSCDv2 for the top 3 m of the soil is shown at the bottom right. These observed distributions are interpolated from a number of discrete soil pedons and therefore have a large associated uncertainty not reflected in these figures. The two different observational data sets have different amounts of soil carbon in the polar region. In the top 2 m of the region mapped by the NCSCDv2 there are 873 Gt C in NCSCDv2 but only 622 Gt C in the WISE30sec data set. The NCSCDv2 was specifically created for the northern high latitudes, so is likely to be more suitable for any assessment of the northern high latitudes soil carbon, but it only covers a limited region of the northern latitudes.

On inspection of Figure 3 the models have more soil carbon in the top 2m than the WISE30sec observations, but this might be expected if the WISE30sec underestimates the northern high latitudes soil carbon. All three models have large amounts of soil carbon in the permafrost regions of Siberia and northern Canada. The right hand column shows the simulated soil carbon top 3m of the simulated permanently frozen soil volume. These are not directly comparable with the NCSCDv2 observations which show the total soil carbon in the top 3 m, the NCSCDv2 observations provide an upper limit on the permafrost carbon (586 Gt C for regions > 60 °N). The permafrost carbon in both JULES simulations falls below this threshold (314 and 488 Gt C for regions > 60 °N), but ORCHIDEE-MICT (959 Gt C for regions > 60 °N) has more than the total soil carbon in NCSCDv2.

The simulated distribution of permafrost carbon is strongly controlled by the simulated permafrost extent: ORCHIDEE-MICT has too much permafrost and hence too much permafrost carbon, JULES



has too little permafrost in north America and western Russia and consequently low permafrost carbon in that region. Although JULES-suppressR$_{esp}$ has suppressed respiration with depth and relatively more soil carbon deeper in the profile, it has a smaller proportion of its total global soil carbon in the northern high latitudes than JULES-deepR$_{esp}$ because of the dependence of $F_T$ on temperature (Equations 2 and 5  3).

Despite obvious model biases, these three different models provide reasonable approximations of the land surface state and we consider then to zero-order as suitable for estimating the permafrost carbon feedback.

## 3.2 Climate projections

10  The simulated areal loss of top 3m or near-surface permafrost under the different RCP scenarios considered is shown in Figure 4. For a grid cell to lose permafrost, it must have a temperature greater than 0ºC at a depth of 3 m for at least one month of the year.  ORCHIDEE-MICT has a much larger initial permafrost extent but loses a smaller fraction of its permafrost than JULES under the RCP scenarios. The models simulate an increasing rate of permafrost loss with time over the next ~100 years, 15  and then tend towards stabilization after 2200 in the RCP scenarios that stabilized forcing around 2100. By 2100 between 5 and 63 % of the permafrost is lost, depending on model configuration and emissions scenario (comparing Figure 4 changes with annotations in Figure1). This potentially very big change in permafrost extent falls within the spread given by Koven et al. (2013) for the CMIP5 models. This might be expected because Koven et al. (2013) found that structural differences in snow physics and 20  soil hydrology had a significant impact on uncertainties - our set of model simulations has a smaller range of these structural uncertainties. Across all scenarios, the near term sensitivity of future permafrost area to global mean temperature change is 1.95 to 2.10 million km$^2$ ºC$^{-1}$ for JULES and 2.30 to 2.55 million km$^2$ ºC$^{-1}$ for ORCHIDEE-MICT. This is less than the 4.0 ± 0.9 million km$^2$ ºC$^{-1}$ found 25  after stabilization of permafrost by Chadburn et al. (2016) but falls within the 1.8 - 2.6 million km$^2$ ºC$^{-1}$ (Chadburn et al., 2015b) found using transient model simulations. By 2300 between 6 and 90 % of the near-surface permafrost is lost, a range more consistent with the stabilized estimate of Chadburn et al.





(2016). In JULES the permafrost area has stabilized by 2300, but ORCHIDEE-MICT is still losing near-surface permafrost, in particular for the RCP8.5 scenario, suggesting that ORCHIDEE-MICT has a larger thermal inertia than JULES.

Figure 5 shows the change in the northern high latitudes vegetation (top row) and soil carbon (middle and bottom) over the region polewards of 60° north. In the case of soil carbon two different quantities are shown - the non-permafrost (non-pf) soil carbon in the middle row and the total soil carbon in the bottom row. At the start of the simulation the non-permafrost soil carbon is defined as the soil carbon within the active layer, i.e. any 'old carbon' below the active layer in the permanently frozen soil is excluded. In any given subsequent year, this non-permafrost soil carbon is defined for the same soil volume, i.e. within the active layer defined for 1860. This non-permafrost soil carbon is taken to be equivalent to the soil carbon assessed by Ito et al. (2015) and Qian et al. (2010) who present results from simulations of the northern high latitudes carbon balance without any specific permafrost carbon included. The bottom row in Figure 5 shows the total northern high latitudes soil carbon including both the 'old carbon' below the active layer and the non-permafrost soil carbon.

Warming and $CO_2$ fertilization effects stimulate vegetation growth and increase land carbon storage in all three land-surface model configurations (Figure 5, top row). This results in an increase of vegetation carbon of between 10 and 60 Gt C by 2100 with a greater increase in ORCHIDEE-MICT than JULES and a greater increase for the higher emissions scenarios, due to higher atmospheric $CO_2$. Ito et al. (2015) used off line land surface models driven by weather data from global climate models under a high emissions scenario and showed the vegetation carbon change was between -5 and 80 Gt C - a much larger spread than found here. Qian et al. (2010) assessed the C4MIP (Coupled Climate Carbon Cycle Model Intercomparison Project) models under a high emissions scenario and found an increasing vegetation carbon of $17 \pm 8$ Gt C by 2100 - this range falls within the spread shown here. The vegetation carbon increase is slower in JULES than ORCHIDEE-MICT and continues to increase after 2300 whereas in ORCHIDEE-MICT the vegetation is stabilizing by 2300. This is probably linked to the





different rates of establishment and growth of the boreal forest as it expands polewards in the two models.

This enhanced vegetation productivity leads to increased soil carbon storage in biomass litter and input to soil organic matter pools. In a warming climate, the soil organic matter decomposition also accelerates, decreasing the soil carbon. The balance between increased soil carbon input and increased decomposition (or reduced turnover time of soil carbon) is relatively uncertain (Jones et al., 2005), leading to simulations of either an increase or decrease in non permafrost soil carbon in the northern high latitudes under future climate change. All three models show an increase in non-permafrost soil carbon before 2100. Non-permafrost soil carbon is defined as any soil carbon that was not in the permanently frozen soil volume at the beginning of the simulation. Across all the different climate responses and emission scenarios examined these increases range from 10 to 100 GtC and suggest that the increase of litter fall dominates over increased respiration. By 2100, Qian et al. (2010) found that the soil carbon in the C4MIP models increases by $21 \pm 16$ Gt C. Ito et al (2015) showed that, although the majority of their model ensemble members have an increase in soil carbon before 2100, there are a few with a decrease. This decrease is not reflected in this ensemble of model simulations and is probably caused by a combination of unsampled structural uncertainty in the current ensemble and unrealistic soil organic carbon distributions in some of the models in the Ito et al. (2015) ensemble. The spread of the future response of the non-permafrost soil carbon in RCP8.5 (caused by differences in the driving GCMs) is larger than the differences between the different RCP scenarios. JULES-suppress$R_{esp}$ and ORCHIDEE-MICT have an increase in non-permafrost soil carbon of similar magnitudes; these increases are slightly larger than in JULES-deep$R_{esp}$. After 2100, in the majority of simulations, the non-permafrost soil carbon is relatively stable. The exception to this is RCP8.5 for JULES-deep$R_{esp}$ where there is a significant loss of non-permafrost soil carbon for a few of the simulations.

Qian et al. (2010) and Ito et al., (2015) did not include any specific permafrost carbon. However when permafrost carbon is included in the simulations (Figure 5, bottom row), the increase in the total soil carbon before 2100 is reduced and in some cases there is a slight decrease. The impact of including



permafrost soil carbon in northern high latitudes soils is highly model dependent. In JULES-suppressR$_{esp}$, although the total soil carbon increases less compared with the non-permafrost soil carbon, there is little noticeable difference in Figure 5. However in ORCHIDEE-MICT and JULES-deepR$_{esp}$ there is a significant decrease in total soil carbon compared with non-permafrost soil carbon,

which continues past 2300 and especially for RCP8.5. In JULES, uncertainties in the total northern high latitude soil carbon (given by the spread in the bottom row of Figure 5) caused by uncertainties in the climate response are larger than the differences between scenarios. However, the differences between the different model versions dominate any differences in scenario or driving climate.

For the ensemble mean of the RCP8.5 scenario, including permafrost carbon in JULES-deepR$_{esp}$ and ORCHIDEE-MICT results in a reduction in the total carbon in the Arctic by 2150 (when compared with 1860). The majority of the RCP2.6 and RCP4.5 scenarios and JULES-suppressR$_{esp}$ still have more total carbon in the Arctic in 2300 when compared with 1860, even when permafrost carbon is included.

### 3.3 Permafrost carbon feedback

Changes in biomass and in global soil carbon drive the land-atmosphere flux of $CO_2$, which then feedbacks influencing the global climate change. IMOGEN can capture this effect. Globally, there is an initial uptake of carbon by the land which reduces over time as the vegetation and soil begins to uptake less, and in some cases the soil becomes a source of carbon as respiration carbon loss overtakes litterfall carbon input (Figure 6 top row). By 2300 the global land surface has a net carbon balance very close to

zero for many of the RCP2.6 and RCP4.5 simulations. The RCP8.5 simulations are very uncertain, with some climate patterns driving a source of global land carbon and some patterns a sink of global land carbon.

The contribution of permafrost carbon to the global land flux is also shown in Figure 6 (bottom row).

Including the permafrost carbon increases the global land $CO_2$ flux to the atmosphere, only slightly for JULES-suppressR$_{esp}$ but more notably for the other two model versions. This brings the time of peak annual uptake earlier in the permafrost enabled simulations - it is 10 years earlier for JULES-deepR$_{esp}$





and ORCHIDEE-MICT and 4 years earlier for JULES-suppressR$_{esp}$ and suggests that permafrost thaw could cause a significant positive feedback on the climate system.

The impact of including these additional permafrost-related carbon fluxes on the global mean temperature is less than ~ 0.46 ºC (Figure 7: PF - non-PF simulations). However, the impact is very different between the three different model configurations, for example, JULES-deepR$_{esp}$ giving an additional increase of 0.02 - 0.28 °C (5$^{th}$ - 95$^{th}$ percentile) and JULES-suppressR$_{esp}$ giving an additional increase of 0.01 - 0.05 °C (5$^{th}$ - 95$^{th}$ percentile). These results appear relatively independent of scenario but there are some notable differences between the different model configurations.

Figure 8 shows the temperature change caused by permafrost carbon loss as a percentage of the global mean temperature change. RCP2.6 has a much lower overall temperature increase (~2 °C) compared with RCP8.5 with a ensemble mean temperature increase of ~7 °C. This is reflected by the larger relative impact of the permafrost carbon for the RCP2.6 scenario compared with the RCP8.5 scenario. For the RCP2.6 scenario the permafrost carbon loss increases the global mean temperature by between 4 and 18 %, however. Even for JULES-suppressR$_{esp}$, where the loss of permafrost carbon is relatively low, the temperature change caused by permafrost carbon is still a relatively large fraction (5 - 8 %) of the global mean temperature change. The percentage impact of permafrost carbon is lower (less than 4% of the global mean temperature change) for the high emissions scenario. This is mainly because a kilogram of $CO_2$ transferred to the atmosphere under a low emissions pathway has a higher radiative efficiency than the same kilogram of $CO_2$ released under a high emissions pathway. These results, in line with MacDougall et al. (2012) and Crichton et al. (2016), suggest that permafrost carbon should be taken into account particularly when evaluating strong mitigation and stabilization scenarios.

### 3.4 Permafrost carbon climate response

The carbon cycle response in a changing world can be described via two components, firstly the climate-carbon response (γ) which determines the change in carbon storage caused by changes in climate. The climate-carbon response, γ, is formally defined as the change in land carbon per degree of global mean temperature change (Friedlingstein et al., 2006). The second component is the concentration-carbon response (β), which determines the change in carbon storage caused by changes

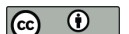

in $CO_2$ concentration - sometimes referred to as the 'fertilization' effect. Chapter 6 of the most recent IPCC report (Ciais et al., 2013) assessed results from models without permafrost carbon and stated that there is high confidence that increasing the atmospheric $CO_2$ will increase land uptake and medium confidence that climate change will reduce the land uptake. That latter can exhibit regional variation, potentially with different signs and is predominantly due to the direct effects of higher temperatures. The inclusion of permafrost carbon will have a minor impact on the concentration-carbon response ($\beta$) but will reduce the land carbon uptake and hence increase the climate-carbon response ($\gamma$).

At the start of the simulation the carbon that is below the active layer is defined as permafrost carbon. In JULES this carbon is numerically labelled and its [depth] location can be traced throughout the simulation. It is denoted 'old permafrost carbon', and is assumed to be the cryogenically stabilized carbon pool within the permafrost under pre-industrial conditions. This can only remain the same or decrease during the simulation period. It cannot be added to. In ORCHIDEE, although the 'old permafrost carbon' can be identified under pre-industrial conditions,

Figure 9 shows the time series of the 'old permafrost carbon' - by 2100 JULES-deep$R_{esp}$ loses between 20 and 50 Gt of old permafrost carbon, and JULES-suppress$R_{esp}$ loses about 20 Gt of old permafrost carbon. There are relatively small differences between emissions scenarios compared with the large differences between JULES-deep$R_{esp}$ and JULES-suppress$R_{esp}$. Koven et al., (2015) found a similar result for RCP4.5 with permafrost soil carbon losses of 12.2–33.4 Gt C, but they found a much larger loss of permafrost carbon for RCP8.5, the high warming scenario. Loss of old permafrost carbon in JULES continues out to 2300, with no sign of stabilization.

Figure 10 shows the change in permafrost carbon as a function of global temperature change for three time slices: 2100, 2200, and 2300. For each time slice and each model version there is a well defined relationship which is relatively independent of the driving climate model and the emissions scenario. The permafrost carbon climate feedback parameter or $\gamma_{PF}$ is defined as the slope of the relationship between the loss of old permafrost carbon and global mean temperature change, i.e. the slope of the



relationship in Figure 10. $\gamma_{PF}$ increases with the time over which the warming has been applied, for example, for JULES-deepR$_{esp}$, $\gamma_{PF}$ is ~10 Gt C / $^o$C at 2100; ~20 Gt C/ $^o$C at 2200 and ~30 Gt C / $^o$C by 2300. These differences are caused by inertia in the permafrost system related to the ongoing low temperatures which slow the decomposition rate of the thawed old permafrost carbon. This significant time dependence of the eprmafrost climate feedback (expressed in Gt C / $^o$C) means that an alternative method of quantifying the permafrost carbon - climate response is required.

**3.5 The Frozen Carbon Vulnerability timescale metrics (FCVt)**

Here we quantify the Frozen Carbon Vulnerability timescale (FCVt), defined for any time over the simulations as the ratio of remaining permafrost carbon to the permafrost carbon loss rate at that time. FCVt can be used to estimate permafrost carbon loss given any pathway of global mean temperature and an assessment of the initial permafrost carbon. It is derived independently for the two different versions of JULES using the old permafrost carbon traced throughout the simulations and the simulated global temperature change. FCVt is defined for any given year as the old permafrost carbon still in the permafrost divided by the loss of permafrost carbon in that year. Figure 11 shows the FCVt as a function of global mean temperature change (GMT) for the two available versions of JULES. There is a clear relationship between the FCVt and the global mean temperature change. This is relatively independent of scenario but remains highly model dependent.

The results of an exponential fit between the FCVt and the global temperature change (Equation 4) are shown in Figure 11 and Table 1. The data for the fit were restricted so that the global temperature change was between 0.2 and 5 °C.

$$FCVt = FCVt_0 \, exp \left(-\frac{\Delta T}{\Gamma}\right) \text{ where } \Delta T > 0.2 \text{ °C} \qquad (4)$$

$FCVt_0$ is a reference timescale representing the permafrost carbon turnover time at the transition point from accumulation of soil carbon to loss of soil carbon. $\Delta T$ is the temperature above which this transition occurs. If permafrost carbon were totally inert, $FCVt_0$ would be infinite at $\Delta T=0$ °C. However in JULES this is a large, but finite number of years and the old permafrost carbon simulated within JULES can be considered stable over centennial timescales. There are a couple of process within JULES which cause this. Firstly, there is mixing of soil carbon throughout the profile. This mixing

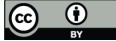

reduces exponentially with increasing depth but still occurs within the permafrost. In addition, the soil carbon is still decomposing, albeit at a very slow rate, at temperatures below zero. $FCVt_0$ is slightly larger for JULES-suppress$R_{esp}$ because the respiration is much slower at depth than in JULES-deep$R_{esp}$. The decay term, $\Gamma$, represents the temperature change at which the number of years taken for all of the old permafrost carbon to be emitted reduces by $1/e$ of its initial value. As expected this is much larger for JULES-suppress$R_{esp}$ than JULES-deep$R_{esp}$.

The relationship found in Equation 4 can be used to reconstruct a simple estimate to quantify the loss of old permafrost carbon given an annual time series of global mean temperature change and the initial permafrost carbon. An example of a reconstructed time series of permafrost carbon is shown in Figure 12. The JULES simulations of old permafrost are the individual curves from Figure 11 for RCP8.5. The reconstructed curves fall within the spread of the original results.

## 4 Conclusions

This paper uses a coupled climate modelling system of intermediate complexity to project additional temperature increases of 0.005 to 0.2 °C by year 2100 and 0.01 to 0.34 °C by year 2300 caused by our projected permafrost carbon feedback. This is in line with previous results (Schuur et al., 2015). A wide range of uncertainties in the future emissions scenario (policy uncertainty); driving climate (spread across GCMs); and parameterisation of the soil carbon decomposition (process uncertainty) are all sampled. The cause of the largest uncertainty is the structural uncertainty in the soil carbon decomposition process. This highlights the need to increase our understanding of the response of permafrost carbon to temperature change in order to constrain future projections.

There are only a limited number of permafrost related processes included within the land surface models. In this example the response of permafrost to climate change is mainly through a deepening of the active layer. However, in many regions of the Arctic there is a high risk of thermokarst related release of carbon. In addition, the results described here only include carbon lost in the form of $CO_2$. There will also be carbon lost as $CH_4$ which will feedback into the atmosphere, although this loss of

$CH_4$ is likely to impact the permafrost carbon feedback less than the release of $CO_2$ (Schadal et al., 2016).

The permafrost carbon feedback has the most significant impact on the mitigation scenario where the temperature change caused by release of permafrost carbon is between 1.5 and 9 % (by 2100) and 6 and 16 % (by 2300) of the global mean temperature change. This has implications for limiting global mean temperature change to 1.5 or 2 degrees where the permafrost carbon feedback should be included in any analysis of these scenarios. We propose a new metric - the Frozen Carbon Vulnerability timescale (FCVt) which can be used to generate the loss of permafrost carbon as a function of global mean temperature change for inclusion into any simple assessment of mitigation scenarios.

## Acknowledgements

The authors acknowledge funding and support from the Permafrost in the Arctic and Global Effects in the 21st century (PAGE21) Framework 7 project GA282700. E.J.B. was supported by the Joint UK BEIS/Defra Met Office Hadley Centre Climate Programme (GA01101) and CRESCENDO (EU project 641816). Chris Huntingford acknowledges the NERC CEH Science Budget. S.E.C. is grateful to the University of Exeter for access to facilities and was supported by the Joint Partnership Initiative project COnstraining Uncertainties in the Permafrost-climate feedback (COUP) (National Environment Research Council grant NE/M01990X/1)

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





| Model | FCV$t_0$ (years; with GMT=0; Equation 4) | $\Gamma$ (°C) | $R^2$ |
|---|---|---|---|
| JULES-deepR$_{esp}$ | 6666 | 2.6 | 0.92 |
| JULES-suppressR$_{esp}$ | 10155 | 4.9 | 0.73 |

**Table 1: Parameters of the exponential fit between permafrost carbon lost per year per remaining permafrost carbon and global mean temperature change.**



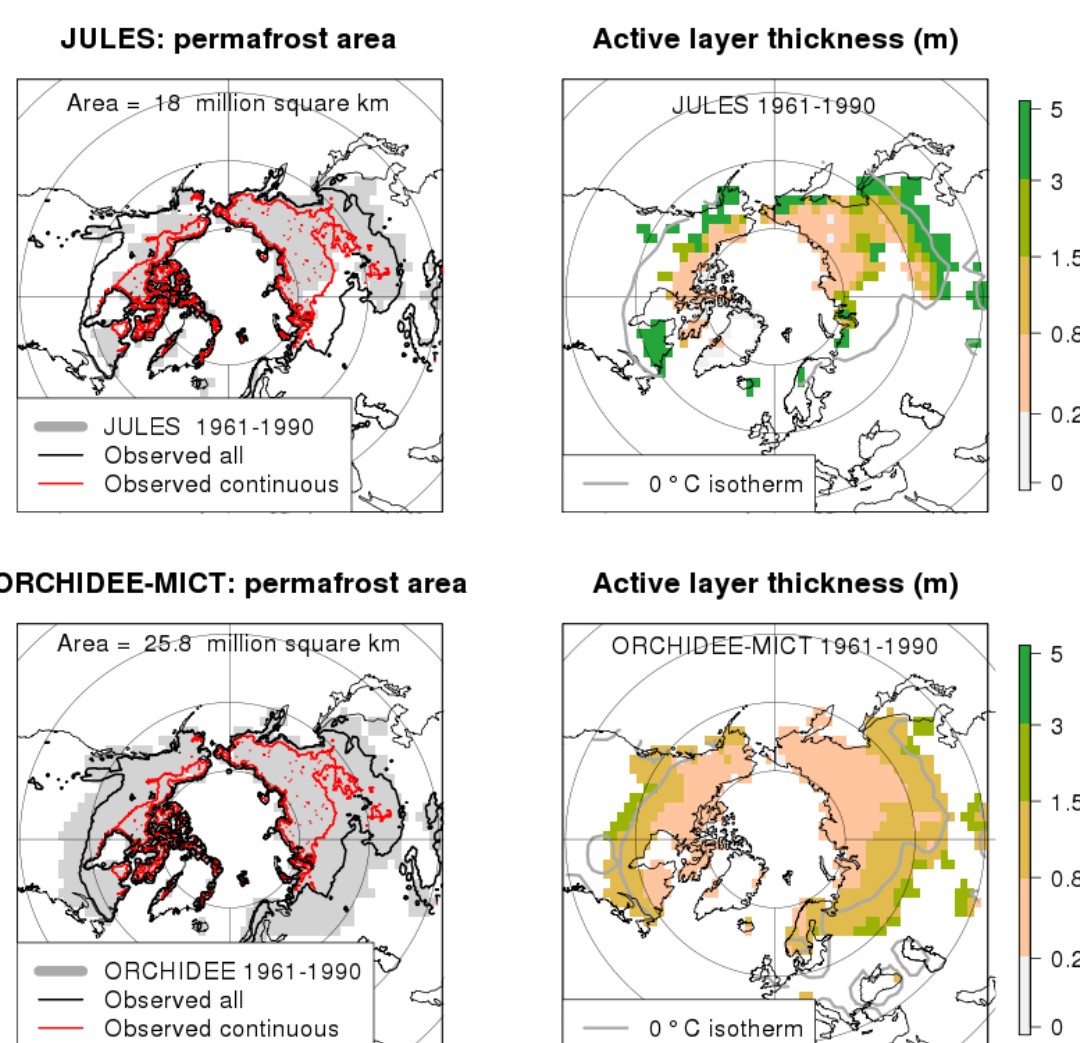

**Figure 1: Simulated permafrost extent (left panels) and maximum summer thaw depth (right panels) for ORCHIDEE-MICT (bottom row) and JULES (top row). Superimposed on the simulated extent is the observed permafrost from Brown et al. (1998). Continuous permafrost is where over 90% of the land surface within the grid cell is underlain by permafrost. The 'All' contour includes regions which have some permafrost present in the grid cell. The zero degree annual mean isotherm from the WATCH 1961-1990 2 m air temperature is drawn on the right hand figures.**



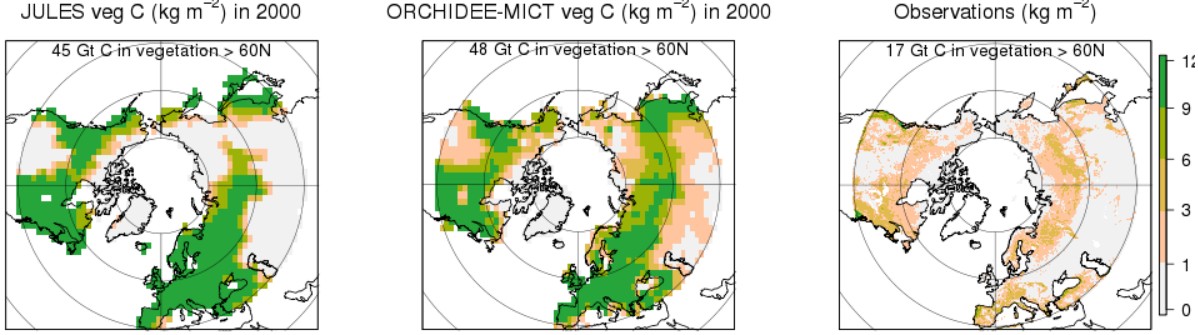

**Figure 2: Simulated vegetation carbon distribution for JULES, ORCHIDEE-MICT for the year 2000 and the observations from the IPCC Tier-1 Global Biomass Carbon Map again for the year 2000 (http://cdiac.ornl.gov/epubs/ndp/global_carbon/carbon_documentation.html).**





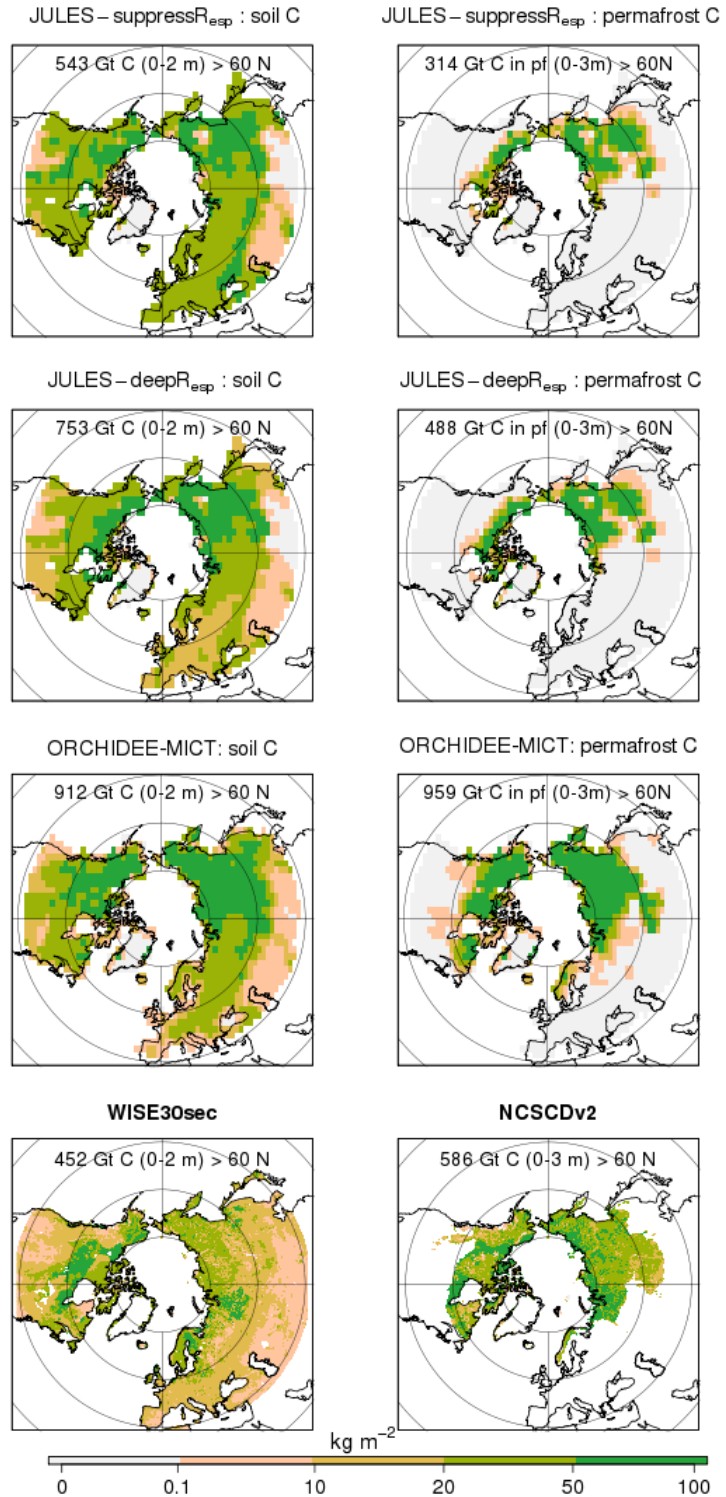





**Figure 3: The simulated distribution of soil carbon in the top 2m (left hand column) and the permafrost carbon in the top 3 m (right hand column) for the three different model versions (first three rows). The bottom row shows the WISE30sec observed global data set for the top 2 m (Batjes, 2016: left hand bottom figure) and the NCSCDv2 northern high latitudes total soil carbon in the top 3 m (Hugelius et al., 2014: right hand bottom figure). In the right hand column the model simulations show just simulated permafrost carbon whilst the NCSCDv2 observations show total soil carbon.**




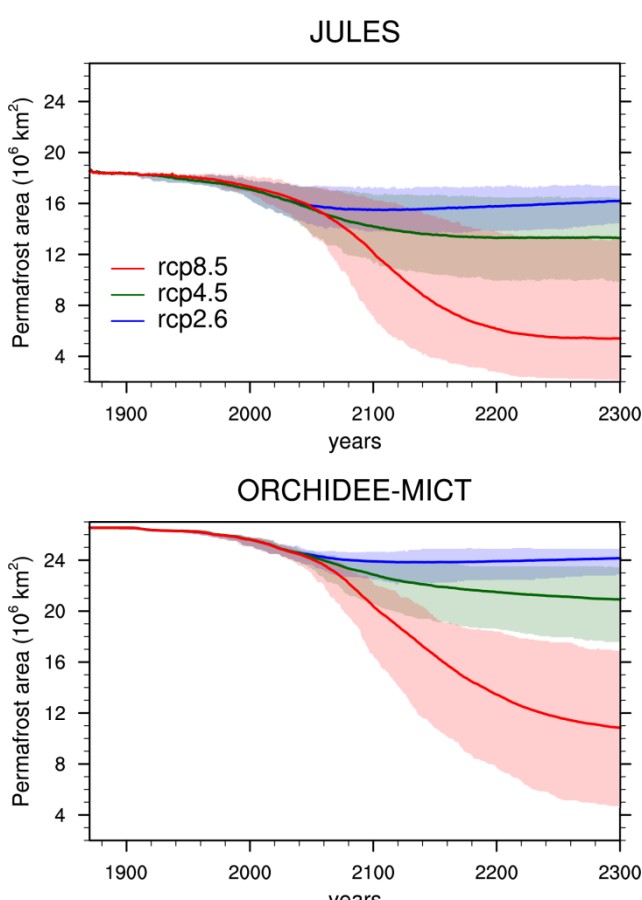

**Figure 4: The areal change in simulated permafrost area extent for the JULES and ORCHIDEE-MICT models, and for three different RCP scenarios. The shaded areas in this and subsequent figures represent the full ensemble spread, accounting for uncertainty in climate response across**
5  **GCMs emulated.**





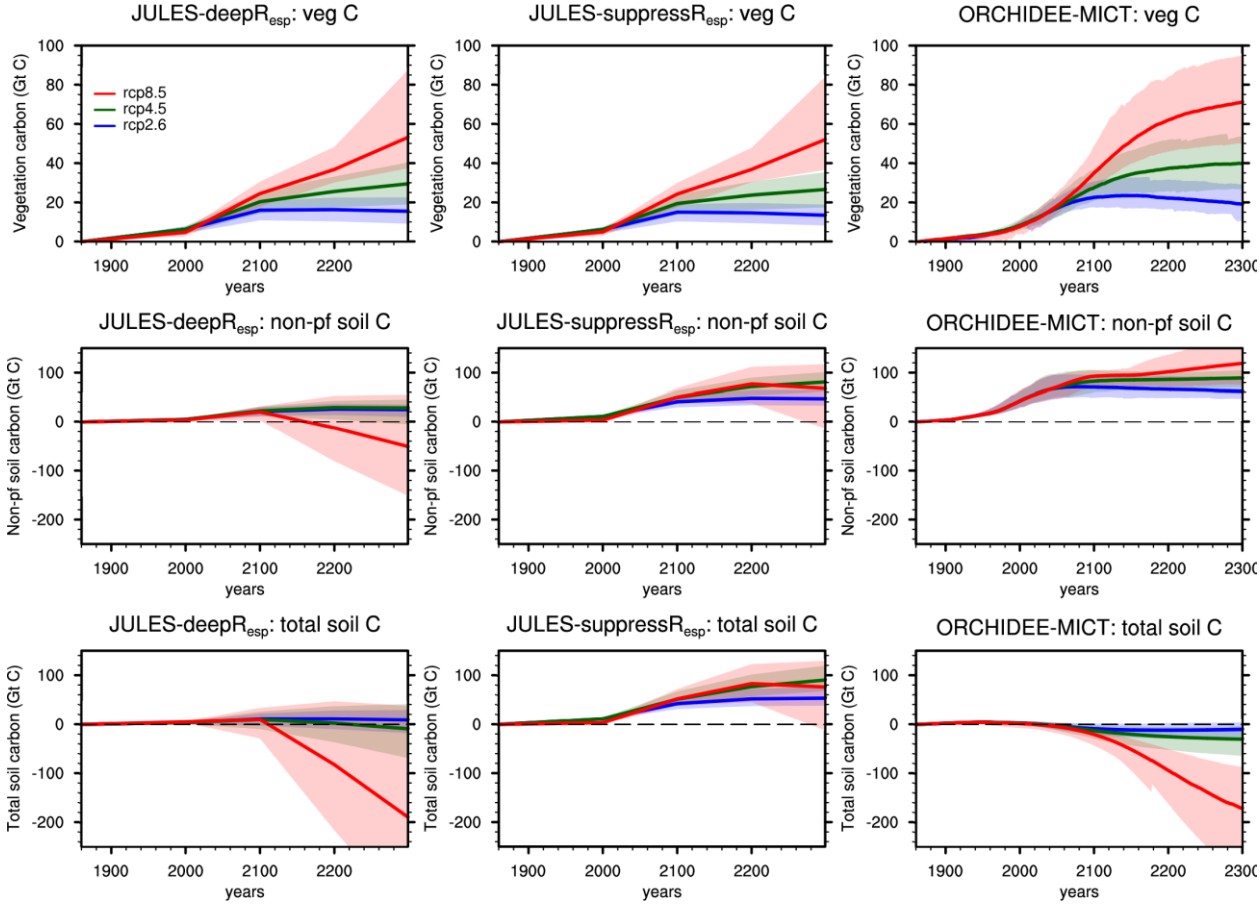

**Figure 5: The change in the vegetation carbon (top row) non-permafrost soil carbon (non-pf; middle row) and total soil carbon (bottom row), all for polewards of 60 degrees north. The vegetation carbon and change is the same in JULES-suppressR$_{esp}$ and JULES-deepR$_{esp}$.**



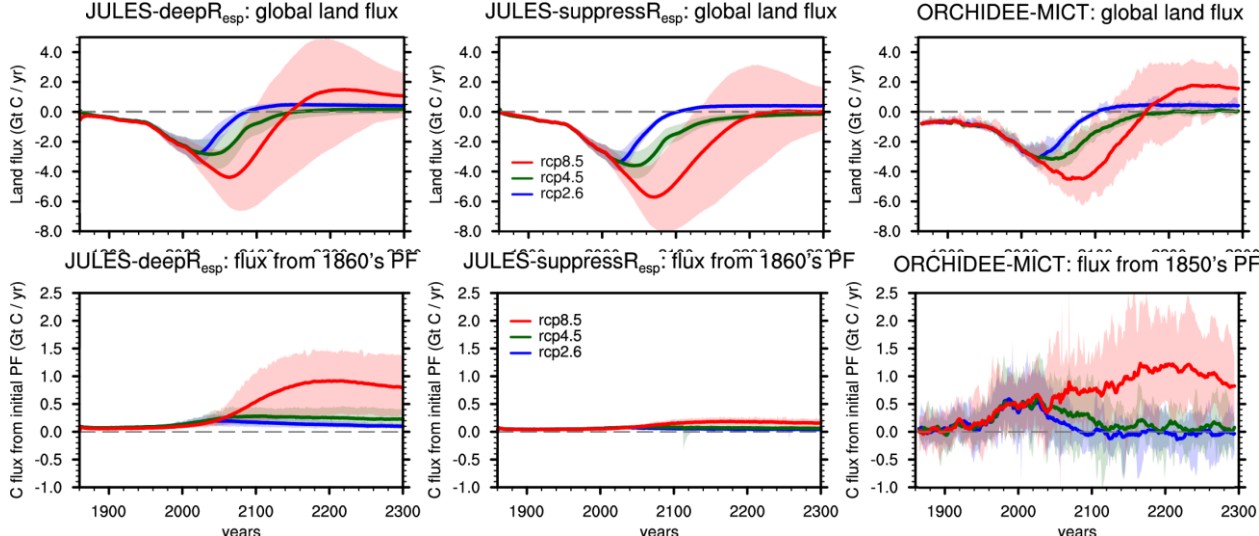

**Figure 6: Global land CO$_2$ flux to the atmosphere (positive is a release to the atmosphere) for the permafrost enabled simulations (PF - top row). The bottom row shows the impact of adding permafrost carbon on the global flux of land carbon to the atmosphere, and its associated feedback via the climate system (difference between PF and non-PF simulations, i.e. PF minus non-PF).**





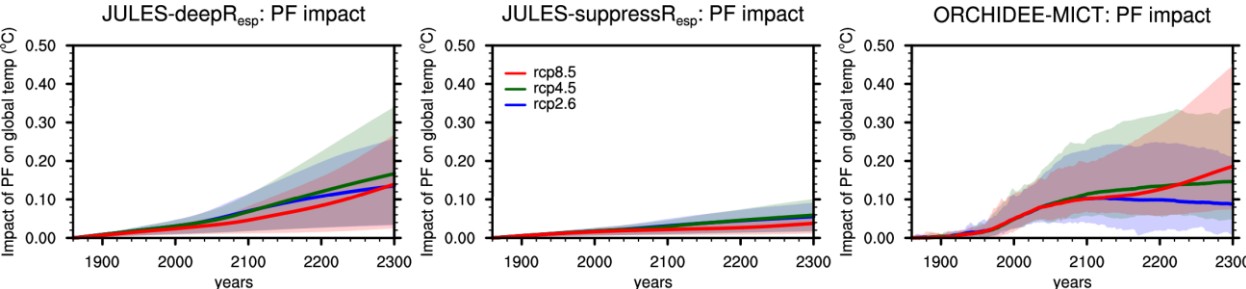

**Figure 7: The impact of the permafrost carbon release on the change in global air temperature (PF - non-PF).**

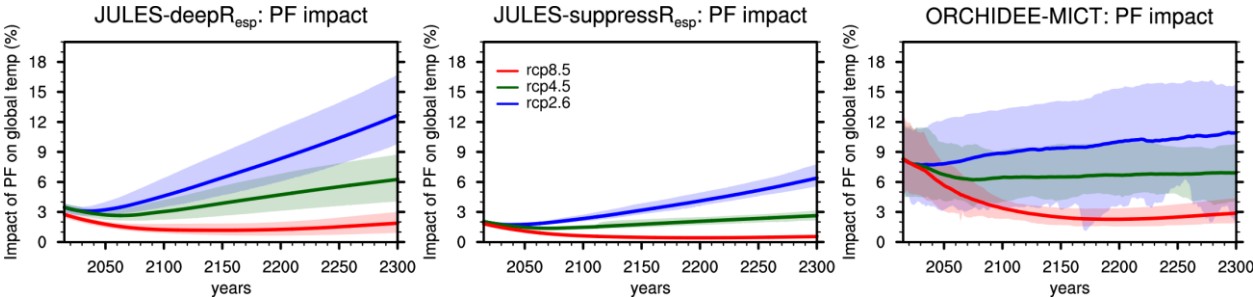

**Figure 8: The percentage impact of the permafrost carbon feedback on the global mean air temperature change (ΔT).**





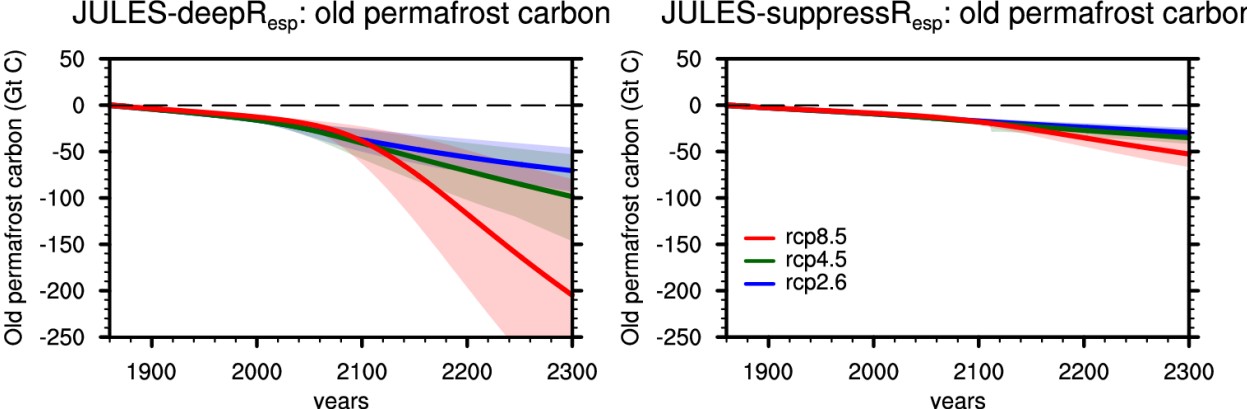

**Figure 9: The change in the old permafrost carbon for JULES. Old permafrost carbon is the labelled carbon identified as being within the permafrost at the start of the simulation.**




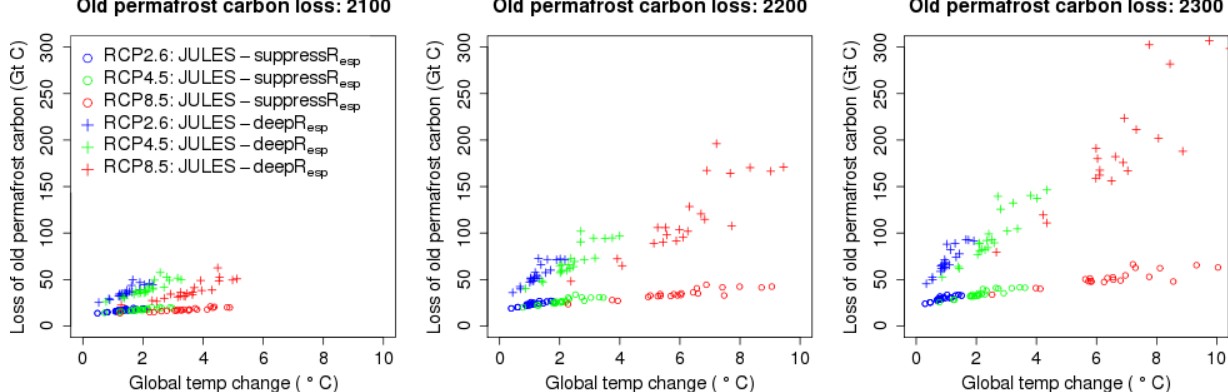

**Figure 10: The relationship between the loss of old carbon from the permafrost region and change in global temperature at years 2100, 2200, and 2300.**





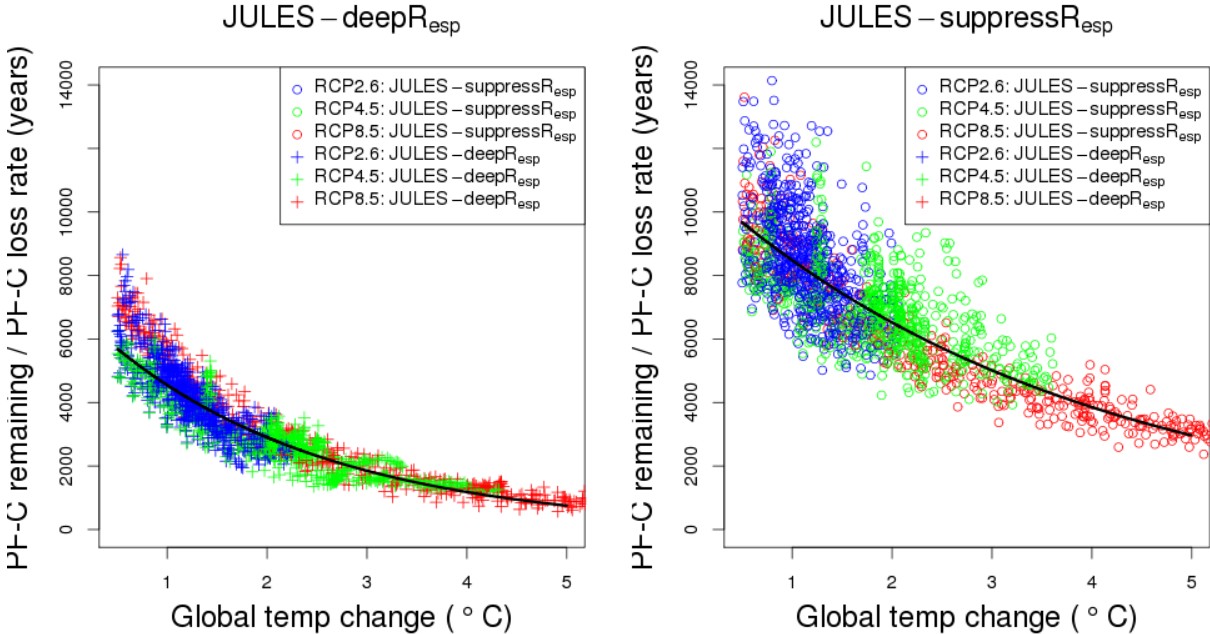

**Figure 11: The permafrost carbon remaining in any given year divided by the loss of permafrost carbon in that year (FCVt) as a function of global mean temperature change (ΔT). The black line is the exponential fit to the model points.**





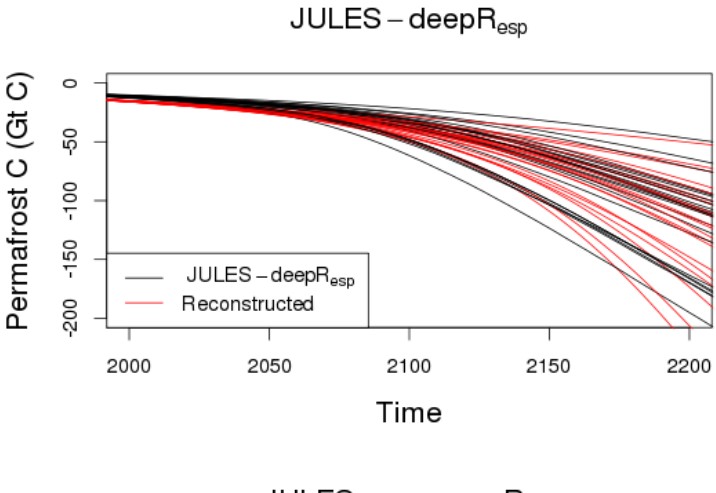

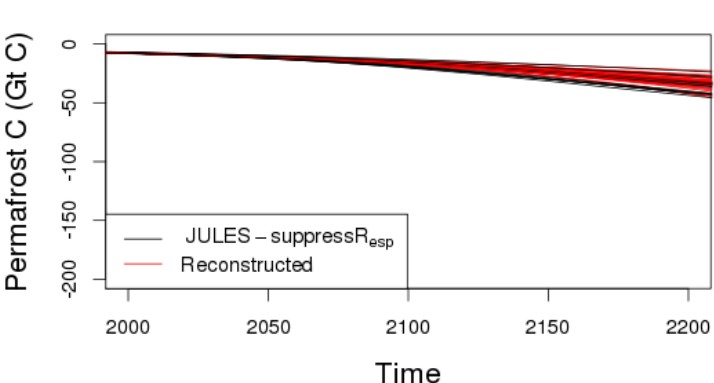

**Figure 12: Time series of permafrost carbon loss for the RCP8.5 scenario. The black lines show the JULES simulations and the red lines show the reconstruction using the initial permafrost carbon, the time series of global mean temperature change and the parameters from Table 1.**