# Peer review of "Quantifying uncertainties of permafrost carbon-climate feedbacks"

_Biogeosciences, 2016_

## Referee Comment (RC1) · Anonymous Referee #1 · 10 Jan 2017

**Review of:** Quantifying uncertainties of permafrost carbon-climate feedbacks

Burke et al.

**Overall evaluation:**

The manuscript documents numerical experiments conducted with three land surface models (two of which are variants of JULES) to estimate the strength of the permafrost carbon cycle feedback to climate change. The experiments show an added warming of 0.2 to 12% by year 2100 and 0.5 to 17% by year 2300 due to the permafrost carbon feedback. The manuscript is well written and provides detailed descriptions of methods used and the results of the experiments.

Overall the manuscript is a timely and original contribution to the study the permafrost carbon feedback to climate change. I recommend that the manuscript be published subject to minor revisions.

**General Comments:**

I am concerned with using $60^oN$ as the boundary for the permafrost zone, given that the continuous permafrost zone extends down to $55^oN$ in Siberia and the discontinuous zone down to $45^oN$. I understand the rational behind selecting common region for comparison for the two land-surface models and the observational data, but leaving so much of the permafrost zone out of the analysis seem misguided. I believe using the region north of $60^oN$ will create problems when comparing model results for future studies.

One solution would be to include results for several regions in a table. For instance: $55^oN$, $60^oN$, the simulated permafrost zone in 1860, and the observed permafrost zone mask. It may also be useful to include both carbon release from the 'old' carbon pool and net change in carbon from all permafrost affected soils.

**Specific Comments:**

**Units:** Gt C $^oC^{-1}$ may be more clear as GtC $K^{-1}$. Having both the C for carbon and the C for Celsius in the same unit can cause confusion.

**Page 2 line 2-4:** This sentence is unclear, please re-write.

**Page 2 line 17:** new-> latest (CMIP5 is now over 4 years old, not that new anymore).

**Page 3 line 14:** I think the symbol here is gamma but if so an odd font is used, and one inconsistent with the gamma used latter in the paper.

**Page 3 line 21:** Citation should be to MacDougall & Knutti (2016).

**Page 4 line 1:** Change "permafrost release of carbon" to "release of permafrost carbon".

**Page 4 line 28:** I cannot find where "GCM" is defined.

**Page 5 1st paragraph:** Include a better rational for using CMIP3 in stead of CMIP5.

**Page 5 line 3:** Change "driven" to "simulated".

**Page 6 line 5:** What is the total depth of the subsurface in the modified JULES?

**Page 7 line 14:** Does the litter C go into permafrost soil layers? If so what stops infinite build-up of C in permafrost layers during spin-up. Is there sub-freezing respiration or does the mixing take care of this?

**Page 8 line 3:** Missing subscript in CO2

**Page 12 line 14:** How do you turn the permafrost carbon off in the model?

**Page 15 line 6 to 9:** Paragraphs need at least two sentences, attached this sentence to the previous paragraph.

**Page 17 line 25:** Somewhere in the manuscript include a caveat on the potential effects of nutrient limitations on the $CO_2$ fertilization effect. It does not seem like either of the models include nutrient limitations thus such limitation may considerably change the results shown here.

**Page 19 line 16:** It may be useful to explain here that the forcing from $CO_2$ is approximately a logarithmic function of $CO_2$ concentration. The way the paragraph is presently worded may through off readers less familiar with radiative forcing.

**Page 21 line 5:** "permafrost" typo.

**Section 3.5:** The quality you define here is a classical residence time (reservoir size over out-flux). Thus "Frozen Carbon Residence Time" may be more intuitive than "vulnerability timescale".

**Page 28 line 1:** Change to MacDougall, A.H. and Knutti, R.
See: doi:10.5194/bg-13-2123-2016

**Table 1:** "GMT" is not defined.

**Figures 1, 3, and 3:** If you continue to use 60oN as you region of intercomparison you should place a line on the maps at the latitude.

**Figure 6:** The captions of Row 2 and of the axis-labels of row 1 overlap.

---

## Referee Comment (RC2) · Anonymous Referee #2 · 31 Jan 2017

Overall Evaluation

This manuscript presents the results of a study that couples three land surface models with vertically stratified soil carbon to an intermediate complexity climate and ocean carbon uptake model to explore climate uncertainties in the context of permafrost carbon-climate feedbacks. The results of this study provide additional information about the impact of permafrost carbon-climate feedbacks in comparison to past studies that have not explored the full range of uncertainty across different climate models. The study also explores a fuller range of uncertainty than has been explored across land surface models. The study provides additional corroboration of the finding of Mac-Dougall et al. (2012) that the permafrost carbon feedback has a greater impact on low emission scenarios than on higher emission scenarios. The study also finds that structural differences among the land surface models are a larger source of uncertainties

than among the climate models. Finally, the study proposes the Frozen Carbon Vulnerability (FCV) metric that can be derived from the simulations to quantify the permafrost carbon response for particular pathways of global temperature change. The FCV may particularly useful to include in integrated assessment methodologies.

In general, I like the design of this study, and the analyses are quite competent. The presentation is very straight forward, and the manuscript is well written. There are of course substantive issues with how the land surface models treat soil carbon, but resolving those issues is beyond the scope of this study which does a very good job of communicating the need to reduce such uncertainties. I think this is a very good first-order study that provides some nice information on the range of uncertainties associated with both structural differences and differences among climate models/scenarios. My comments below are all minor.

Specific Comments

Page 1, line 28: Change "this range reflecting" to "these ranges reflecting".

Page 2, line 3: Change "between climate models" to "among climate models".

Page 2, line 5: Change "is dependent on" to "depends on".

Page 2, line 20: Change "large stocks . . . is stabilised" to "large stocks . . . are stabilised".

Page 2, line 22: Change "Adding the" to "The addition of" to avoid dangling participle.

Page 3, line 24: Change "theory" to "approach".

Page 4, line 1: Change "0.1 and 0.8" to "0.1 to 0.8".

Page 8, lines 1-4: It seems to me that the last two sentences of section 2.1 are redundant.

Page 9, line 26: Change "is itself dependent" to "itself depends"?

Page 11, line 6: Change "spun up" to "spin up"?

Page 12, line 1: Change "10 times each time followed by 2 year run of the full" to "10 times, with each time followed by a 2-year run of the full"?

Page 12, line 9: Change "In order to" to "To".

Page 12, line 20: Change "in order to" to "to".

Page 15, line 1: Change "north America" to "North America".

Page 18, lines 12 and 13: I think this is the first use of "Arctic" in the manuscript. I suggest change "Arctic", which has several connotations and has not been defined in the manuscript, to "northern permafrost region".

Page 21, line 5: Change "eprmafrost" to "permafrost".

Page 22, line 21: Change "in order to" to "to".

Page 23, line 1: Correct spelling of "Schadal".

---

## Referee Comment (RC3) · Anonymous Referee #3 · 4 Feb 2017

Overall impression

In the current study, Burke et al. have used three versions of land surface schemes coupled to a climate-carbon model of intermediate complexity for investigating the contribution of permafrost carbon to global warming under various anthropogenic emission scenarios. The authors explore uncertainty in their estimates by considering a spread in climate forcings, and by accounting for structural model uncertainty regarding the description of soil respiration. Further, the authors derive a new metric (of interest to integrated assessment studies), which quantifies the permafrost carbon response (the Frozen Carbon Vulnerability timescale) – independent of the pathway of global temperature change. The manuscript is well structured and presented, while the simulation experiments follow a clear design and convincing strategy.

[Figure]

General comments

- The authors consider uncertainty in climate forcings and in structural uncertainty concerning simulated respiration rates. I am missing a discussion of further factors which are subject to uncertainty and likely affect the model outcomes. Amongst others, there is e.g. large uncertainty in vertical soil organic carbon (SOC) distribution, in partitioning of organic matter into different lability classes, in assumed respiration rates. Further, a large portion of SOC resides in organic rich deposits (histels) with different environmental controls compared to mineral soils. How do the authors deal with this issue? Implications of not explicitly accounting for these deposits should be discussed in the text.

- The model simulations illustrate the dominant control of the assumption concerning soil respiration (difference between suppress and deep respiration). Is there any evidence which of the schemes is more likely to approximate "reality"? Commenting on this issue would be helpful for strategies of reducing uncertainty in future simulations.

- Regional definition of fluxes summed over the region polewards of 60° north It is unclear to what extent the results are biased by contributions from non-permafrost regions, and what is missed from regions of permafrost south of 60°N. See also comments and suggestions from referee 1.

Specific comments

– Why is the cryoturbation mixing rate not chosen a function of active layer thickness (instead of choosing a fixed value of 3 meters?). Assuming that the effect of cryoturbation would be largely felt in the active layer, the discussed scheme seems to overestimate transport by cryoturbation to depth for shallow active layer sites. How do simulated vertical SOC profiles compare with data? A few representative sites could (very generally) be discussed. - "The data for the fit were restricted so that the global temperature change was between 0.2 and 5 °C." How do results look like for T>5°C? If the fit gets rather poor, this should be made clear to avoid a mis-use of the functional

dependency for high-end emission scenarios when applied for integrated assessment studies.

- A simple figure in the SI to graphically show the functional forms of equation 2 & 3 would be helpful

- "The range of climate sensitivity and regional distribution of climate change in the CMIP3 models is comparable with that in the CMIP5 models.] "

Would be good to indicate the range of considered climate sensitivities in this study here.

---

## Referee Comment (RC4) · Anonymous Referee #4 · 8 Feb 2017

Overall Evaluation:

The manuscript by Burke et al. quantifies uncertainties of the permafrost carbon – climate feedbacks from three perspectives: future emission scenarios, different climate sensitivities and regional climate change patterns, and land surface models with different land surface processes. The paper addresses a timely and important topic, which fits the need to increase our understanding of permafrost carbon – climate feedbacks for the aim to constrain future climate projections. In the work, three versions of land surface models were coupled to a intermediate complexity climate and ocean carbon uptake model. The study showed the ranges of additional temperature increases caused by permafrost carbon feedback by year 2100 and 2300. The different representations of land surface processes, especially about soil carbon decomposition, in land surface models can introduce bigger uncertainties than climate models in estimating the permafrost carbon – climate feedbacks. The response of permafrost carbon to climate is found to be dependent on the temporal trajectory of warming as well as the absolute amount of warming. Finally, the authors proposed a new policy relevant metric which is considered to be useful for integrated assessment of mitigation scenarios. Those results are meaningful and can serve as a valuable reference for related work. In general, in my opinion, the study is competent and indicates original contribution in the experiment design. The results were thoroughly analyzed and the conclusions are significant. The manuscript is written well with a clear structure and fluent language.

I only have a few specific comments for the authors to consider:

- Page 1, Line 21: Change " Simulations were performed" to "Those simulations ..."

- Page 2, Line 7: "more complex land surface models", more compare to what?

- Page 2, Line 20-21: please rephrase this sentence.

- Page 6, Line 25: please mention the names of the two different parameterisations right there in a parenthesis after "Two different parameterisations".

- Page 12, Line 10: should the "two ORCHIDEE" be "the ORCHIDEE"?

- Page 12, Line 12-14: I feel the definition of permafrost carbon is not necessarily to be given here because it has been given in the very beginning in Page 2, Line 20 and also in Page 7, Line 27. Maybe you can just simply write here that the permafrost carbon refers to the old permafrost carbon.

- Page 13, Line 9: give the definition of continuous permafrost here in parenthesis.

- Page 13, Line 10: should the discontinuous permafrost be defined as more than 50% but less than 90% of a grid cell underlain by permafrost?

- Page 17, Line 10-11: Again, the definition of non-permafrost soil carbon has been given in Page 16, Line 8-10. Perhaps it is not necessary to be given here again.

[Figure]

- Page 18, Line 2: "Less", should here be "more" according to the figure.

- Page 20, Line 14, the last sentence is not finished . . .

- Page 36, the figure caption: Change "the vegetation carbon and change is" to "the vegetation carbon and its change are".

---

## Author Comment (AC1) · 22 Mar 2017

**Quantifying uncertainties of permafrost carbon-climate feedbacks – response to reviewers' comments.**

**Reviewer 1**

The reviewer has concerns about using 60 degrees north as a "boundary" for the permafrost zone and its relevance for future studies. Ironically this was chosen because two previous studies (Ito et al. 2015; and Qian et al., 2010) used these boundaries! However, I agree this is rather limited for permafrost studies as it is missing regions where there is permafrost. A table will be added which includes the vegetation carbon, total soil carbon and non permafrost soil carbon for three regions > 55 degrees N, the observed permafrost extent and the 1860 simulated permafrost extent. In each case the 'mean (5-95$^{th}$ percentile)' of the change in carbon will be shown for 2100 and 2300

All specific comments will be addressed. Of particular note:

*Page 5, paragraph 1 – include a better rational for using CMIP3 instead of CMIP5.*
The main reason for using CMIP3 vs CMIP5 is that the CMIP5 patterns were not available to drive the simulations. However, given that the range of climate sensitivity and regional distribution of climate change are very similar in both model ensembles this will not alter the nature of the results. All future simulations will be carried out using the CMIP5 patterns.

*Page 6, line 5:*
The total depth of the subsurface in the modified JULES is 18.3 m. This has been added.

*Page 7, line 14 – does the litter C go into the permafrost soil layers, if so what stops the build up of C in permafrost layers during spin up?*
The litter carbon may well go into the permafrost layers during spin-up. However, there is both subfreezing soil respiration and mixing both of which stop infinite build up of carbon in the permafrost layers. A new figure has been added which shows the response of the soil respiration to temperature for the three different model versions along with the following sentence: "All three functions have some decomposition at temperatures below freezing". The following sentence has also been added to the document: "The litter is mixed through the soil profile by either bioturbation or cryoturbation"

*Page 12, line 14: How do you turn the permafrost carbon off in the model?*
"In JULES the permafrost carbon and non-permafrost carbon are diagnosed separately at each timestep. For the non-PF case, only the non-permafrost carbon is visible to IMOGEN, whereas for the PF simulation all the soil carbon is visible to IMOGEN. In ORCHIDEE-MICT, for the case of the non-PF simulations, the pre-industrial permafrost carbon is subtracted from the total soil carbon at each timestep" This has been clarified in the text.

*Page 17 line 25: Nutrient limitation is not included and may change these results – please can you comment.*
A caveat about the potential affect of nutrient limitation is included in the conclusions.

*Page 19 line 16: It may be useful to explain here that the forcing from CO2 is approximately a logarithmic function of CO2 concentration.*
The following has been added: "This is because the radiative forcing from $CO_2$ is a logarithmic function of $CO_2$ concentration – at higher $CO_2$ concentrations, 1 kg of $CO_2$ increases the radiative forcing less than at lower concentrations."

*Section 3.5: The quality you define here is a classical residence time (reservoir size over out-flux). Thus "Frozen Carbon Residence Time" may be more intuitive than "vulnerability timescale".*
This has been changed to Frozen Carbon Residence time (FCRt) as suggested.

*Figures 1, 2, and 3: If you continue to use 60oN as your region of intercomparison you should place a line on the maps at that latitude.*
I have added these lines.

**Reviewer 2**

This reviewer notes that "comments are minor" - these are all syntax changes and have been made according to the suggestions by the reviewer.

**Reviewer 3**

*The authors consider uncertainty in climate forcings and in structural uncertainty concerning simulated respiration rates. I am missing a discussion of further factors which are subject to uncertainty and likely affect the model outcomes. Amongst others, there is e.g. large uncertainty in vertical soil organic carbon (SOC) distribution, in partitioning of organic matter into different lability classes, in assumed respiration rates. Further, a large portion of SOC resides in organic rich deposits (histels) with different environmental controls compared to mineral soils. How do the authors deal with this issue? Implications of not explicitly accounting for these deposits should be discussed in the text.*
A discussion of these factors was added to the second paragraph of the conclusions. These are structural uncertainties which have not yet been evaluated. This paragraph now reads as follows: "There are only a limited number of permafrost related processes included within the land surface models. In this example the physical response of permafrost to climate change is mainly through a deepening of the active layer. However, in many regions of the northern permafrost region there is a high risk of thermokarst a process not included in the models. The model structural uncertainty is based around differences in the response of the respiration to temperature. However, there are additional biogeochemical structural model uncertainties such as the partitioning of organic matter into different lability pools along with their turnover times and the dependence of decomposition on moisture, including any differences in these processes between organic rich and mineral soils. In addition, the results described here only include carbon lost in the form of $CO_2$. There will also be carbon lost as $CH_4$ which will feedback into the atmosphere, although this loss of $CH_4$ is likely to impact the permafrost carbon feedback less than the release of $CO_2$ (Schadel et al., 2016)."

*The model simulations illustrate the dominant control of the assumption concerning soil respiration (difference between suppress and deep respiration). Is there any evidence which of the schemes is more likely to approximate "reality"? Commenting on this issue would be helpful for strategies of reducing uncertainty in future simulations.*

This is a very interesting question and one the authors would like to address in future work. One way of assessing this is to look at the residence times of the soil carbon as a function of depth (e.g. Schädel, C., Schuur, E.A., Bracho, R., Elberling, B., Knoblauch, C., Lee, H., Luo, Y., Shaver, G.R. and Turetsky, M.R., 2014. Circumpolar assessment of permafrost C quality and its vulnerability over time using long-term incubation data. Global change biology, 20(2), pp.641-652.). An alternative method is to design field studies to monitor the profile of soil respiration within the soil. The following sentence from the conclusions has been expanded: "This highlights the need to increase our understanding of the response of permafrost carbon to temperature change to constrain future projections by utilising observations of, for example, the depth dependence of the soil carbon residence time or the soil respiration.

*Regional definition of fluxes summed over the region polewards of 60◦ north It is unclear to what extent the results are biased by contributions from non-permafrost regions, and what is missed from regions of permafrost south of 60◦ N. See also comments and suggestions from referee 1.*
As suggested above a table has been added to the document using other regional definitions.

*Why is the cryoturbation mixing rate not chosen a function of active layer thickness (instead of choosing a fixed value of 3 meters?). Assuming that the effect of cryoturbation would be largely felt in the active layer, the discussed scheme seems to overestimate transport by cryoturbation to depth for shallow active layer sites.*
This is a very useful suggestion and should be incorporated as a further model development.

*How do simulated vertical SOC profiles compare with data? A few representative sites could (very generally) be discussed.*
Burke et al. (2017) (http://www.geosci-model-dev.net/10/959/2017/gmd-10-959-2017.pdf) shows the profile of soil carbon in the permafrost soils in Figure 7. The total amount of soil carbon is very dependent on the litterfall. In general, for mineral soils the shape of the profile is in relatively good agreement with the observations, but there are discrepancies for peat soils caused by peat forming processes missing from the model. There is a forthcoming paper discussing the results from JULES simulations for a small number of Arctic sites which assesses this in greater detail. The following has been added to the paper discussing this: "Using pan-arctic JULES simulations with this vertically resolved soil carbon model Burke et al. (2017) showed that, at the large scale, the depth distribution of soil organic carbon approximately follows that of the observations. Chadburn et al. (2017) suggests that, given the correct input (litter), the depth distribution of soil organic carbon is well simulated for mineral soils, but that the model is currently unable to reproduce the peat layers of organic soils."

*"The data for the fit were restricted so that the global temperature change was between 0.2 and 5 ◦ C." How do results look like for T>5◦ C? If the fit gets rather poor, this should be made clear to avoid a mis-use of the functional scenarios when applied for integrated assessment studies.*
A caveat stating that these relationships shouldn't be used for temperature changes greater than 5 degrees has been added.

*A simple figure in the SI to graphically show the functional forms of equation 2 & 3 would be helpful.*
This figure is shown as Figure 1 in Burke et al. (2017) and ([http://www.geosci-model-dev.net/10/959/2017/gmd-10-959-2017.pdf](http://www.geosci-model-dev.net/10/959/2017/gmd-10-959-2017.pdf)) is referred to.

*"The range of climate sensitivity and regional distribution of climate change in the CMIP3 models is comparable with that in the CMIP5 models". Would be good to indicate the range of considered climate sensitivities in this study here.*
This has been added to the paper.

**Reviewer 4**

All minor syntax modifications recommended have been changed. The following changes have also been made:

*- Page 2, Line 7: "more complex land surface models", more compare to what?*
"more" has now been removed.

*- Page 12, Line 12-14: I feel the definition of permafrost carbon is not necessarily to be given here because it has been given in the very beginning in Page 2, Line 20 and also in Page 7, Line 27. Maybe you can just simply write here that the permafrost carbon refers to the old permafrost carbon.*
This has been removed from the document.

*- Page 13, Line 9: give the definition of continuous permafrost here in parenthesis and Page 13, Line 10: should the discontinuous permafrost be defined as more than 50% but less than 90% of a grid cell underlain by permafrost?*
These have been changed – continuous is defined as > 90 % and discontinuous as between 50 and 90 %.

*- Page 17, Line 10-11: Again, the definition of non-permafrost soil carbon has been given in Page 16, Line 8-10. Perhaps it is not necessary to be given here again.*
This additional definition of permafrost carbon has been removed.

*- Page 18, Line 2: "Less", should here be "more" according to the figure.*
This has been changed.

*- Page 20, Line 14, the last sentence is not finished . . .*
This sentence is now finished: "In ORCHIDEE although the 'old permafrost carbon' can be identified under pre-industrial conditions, it cannot be traced throughout the simulations."